# QuantumBoost: A lazy, yet fast, quantum algorithm for learning with weak hypotheses

Amira Abbas [1]   Yanlin Chen [2]   Tuyen Nguyen [3]   Ronald de Wolf [4]

## Abstract

The technique of combining multiple votes to enhance the quality of a decision is the core of boosting algorithms in machine learning. In particular, boosting provably increases decision quality by combining multiple "weak learners"—hypotheses that are only slightly better than random guessing—into a single "strong learner" that classifies data well. There exist various versions of boosting algorithms, which we improve upon through the introduction of QuantumBoost. Inspired by classical work by Barak, Hardt and Kale, our QuantumBoost algorithm achieves the best known runtime over other boosting methods through two innovations. First, it uses a quantum algorithm to compute approximate Bregman projections faster. Second, it combines this with a lazy projection strategy, a technique from convex optimization where projections are performed infrequently rather than every iteration. To our knowledge, QuantumBoost is the first algorithm, classical or quantum, to successfully adopt a lazy projection strategy in the context of boosting.

## 1. Introduction

With its simplicity and provable efficiency, boosting is one of the few algorithmic frameworks in machine learning that is both well-understood theoretically and widely used in practice. The idea was first posed by Kearns and Valiant (Kearns & Valiant, 1994) in the context of probably approximately correct (PAC) learning (Valiant, 1984). They

conjectured the ability to "boost" a weak learning algorithm, which performs slightly better than random guessing, into a strong learning algorithm with an arbitrarily small generalization error. Schapire (Schapire, 1990) was the first to introduce such a provably polynomial-time boosting algorithm, followed by Freund (Freund, 1995) who further improved the efficiency – although practical bottlenecks still persisted. These bottlenecks were alleviated through the introduction of AdaBoost, a remarkable algorithm created by Freund and Schapire (Freund & Schapire, 1997) which, till this day, remains competitive for various machine learning tasks (Arora et al., 2012; Viola & Jones, 2001; Drucker et al., 1999; Freund & Schapire, 1996). For illustration, consider the canonical task of binary classification. We start from a training set $S = \{(x_i, y_i)\}_{i=1}^{m}$ where each $x_i \in \mathcal{X}$ is distributed according to some (possibly unknown) distribution $D$, and labeled with a $y_i \in \mathcal{Y}$. These labels could, for instance, be chosen according to some unknown target function $f : \mathcal{X} \to \mathcal{Y}$ that we want to learn. For simplicity, let $\mathcal{X} = \{0, 1\}^n$ and $\mathcal{Y} = \{-1, 1\}$.[1] A weak learner is typically fed examples from the set $S$ according to some distribution that we ourselves choose, say $D'$, and is promised to output a hypothesis $h : \{0, 1\}^n \to \{0, 1\}$ that performs slightly better (under $D'$) than random guessing:

$$\Pr_{x_i \sim D'}[h(x_i) = y_i] \geq \frac{1}{2} + \gamma, \qquad (1)$$

where $\gamma \in (0, 1/2)$ denotes the strength of the weak learner, meaning its advantage over random guessing. The key idea of AdaBoost is to start with a $D'$ that is uniform over the training set, and to call the weak learner over a series of iterations $t = 1, \ldots, T$, each time updating $D'$ to force the learner to focus on examples that are frequently misclassified by the hypotheses produced in earlier iterations. Using AdaBoost's update strategy and particular method of combining the weak learner's $T$ hypotheses into one final hypothesis $H$, Freund and Schapire (Freund & Schapire, 1997) showed that $T = O(\log(1/\epsilon)/\gamma^2)$ iterations suffice to achieve low *empirical error*:

$$\Pr_{i \sim [m]}[H(x_i) \neq y_i] \leq \epsilon, \qquad (2)$$

[1]Google Quantum AI, Venice, California, 90291, USA.
[2]University of Maryland, College Park, MD 20742, USA.
[3]University of Technology Sydney, Ultimo, NSW, Australia.
[4]QuSoft, CWI and University of Amsterdam, the Netherlands. Partially supported by the Dutch Research Council (NWO) through Gravitation-grant Quantum Software Consortium, 024.003.037. Correspondence to: Amira Abbas <amiraabbas@google.com>, Yanlin Chen <yanlin@umd.edu>.

*Proceedings of the 43rd International Conference on Machine Learning*, Seoul, South Korea. PMLR 306, 2026. Copyright 2026 by the author(s).

---

[1]The assumption that $\mathcal{X}$ is the Boolean cube and that the labels (function values) are binary, is not necessary for boosting to work. We could of course also use $\mathcal{Y} = \{0, 1\}$ as a range for $f$.

where the notation "$i \sim [m]$" means that $i$ ranges uniformly over $1, \ldots, m$. If the labels $y_i$ correspond to a target function $f$, then VC-theory implies that, for large enough sample size $m$, this low empirical error actually implies low *generalization error* w.r.t. the data-generating distribution $D$:

$$\Pr_{x \sim D}[H(x) \neq f(x)] \leq \epsilon', \tag{3}$$

for an $\epsilon'$ that is only slightly bigger than $\epsilon$. Generalization error measures error over the whole domain of $f$ (weighted by $D$), not just the $x_i$'s that happened to be part of the training set. In other words, we have learned a good approximation of $f$ that generalizes well beyond the training set.

In an attempt to accommodate more realistic learning scenarios, Servedio (Servedio, 2003) introduced a boosting algorithm called SmoothBoost which allows for learning in the presence of a small amount of malicious noise, but uses $T = O(1/(\epsilon\gamma^2))$ iterations, which has a much worse $\epsilon$-dependence than AdaBoost. The main idea of Smooth-Boost is to ensure that the updated distributions remain "smooth" at every iteration, meaning the weight assigned to each example is not much larger than uniform probability $1/m$. As long as $T \ll m$, this smoothness property prevents a few (possibly malicious) errors in the labels of the $m$ examples from having undue influence over the final hypothesis. Interestingly, a variant of SmoothBoost was developed by Kale (Kale, 2007) to construct "hard-core sets", a form of hardness amplification of Boolean functions. The connection between hard-core set construction and boosting in learning theory is rather elegant: boosting methods hone in on examples that are difficult for a learner to classify, implying the existence of a so-called hard-core set (a set of inputs that are hard to classify). Kale's SmoothBoost algorithm gives a size matching the best known parameters of other hardcore-set constructions (Klivans & Servedio, 1999; Impagliazzo, 1995). Additionally, Kale's Smooth-Boost matches the favorable number of iterations in AdaBoost, with $T = O(\log(1/\epsilon)/\gamma^2)$, yielding a practical algorithm for learning in the presence of malicious noise.

This paper is about *quantum* algorithms for boosting. Thus far, all classical boosting techniques call the weak learner at each iteration and update an explicit weight vector over the $m$ examples in the training set. Denoting the runtime associated with calling the weak learner as $W$ (which we also use as an upper bound for the number of examples that a run of the weak learner needs), this inevitably incurs a runtime at least linear in $W$ and $m$ at each iteration of the algorithm. In an attempt to improve this scaling, AdaBoost was first quantized in (Arunachalam & Maity, 2020) and subsequently, SmoothBoost was quantized in (Izdebski & de Wolf, 2023). These quantum boosting algorithms both quadratically improve the runtime scaling in $m$, at the expense of a scaling in $\gamma$ that is worse than their classical counterparts.

## 1.1. Our results

We introduce a new quantum algorithm for boosting that we call QuantumBoost, based on quantizing Kale's version of SmoothBoost. It removes the explicit dependence on $m$ and matches AdaBoost's scaling in $\gamma$—something other existing quantum proposals for boosting did not achieve. Moreover, QuantumBoost is the first boosting method (classical or quantum) to improve the runtime dependence on $1/\epsilon$ to $\widetilde{O}(1/\sqrt{\epsilon})$. Our contributions can be stated as follows.

**Theorem 1.1** (Informal: Empirical error and runtime of QuantumBoost). *Given access to a $\gamma$-weak learner for the concept class $\mathcal{C}$ with hypothesis class $\mathcal{H}$ and runtime $W$, and a training set $S = \{(x_i, y_i)\}_{i=1}^m$, QuantumBoost produces a hypothesis with empirical error (on the training set $S$) that is at most $\epsilon$, with an overall runtime of*

$$\widetilde{O}\left(\frac{W}{\sqrt{\epsilon}\gamma^4}\right). \tag{4}$$

**Theorem 1.2** (Informal: Generalization error of Quantum-Boost). *Let $d$ be the VC-dimension of the hypothesis class $\mathcal{H}$ from which the $\gamma$-weak learner produces hypotheses. For a sufficiently large training set size*

$$m = \Theta\left(\frac{d\log(d/\epsilon)}{\gamma^2\epsilon^2} + \frac{\log(1/\delta)}{\epsilon^2}\right), \tag{5}$$

*for every target function $f \in \mathcal{C}$, for every data-generating distribution $D$, with success probability $1 - \delta$, Quantum-Boost produces a hypothesis with generalization error $\leq \epsilon$.*

Our improvements over earlier boosting algorithms essentially come from three sources:

1. It was noted in (Barak et al., 2009) that for Kale's version of SmoothBoost, approximate Bregman projections can be used to project measures onto the set of so-called high-density measures. These high-density measures, when normalized, are smooth distributions. This avoids the need to explicitly verify smoothness by merely projecting onto the set of high-density measures. We quantize Kale's SmoothBoost algorithm and show how such an (approximate) projection can be computed more efficiently with quantum techniques.

2. We adopt a lazy projection strategy that only projects at every $K^{\text{th}}$ iteration, where $K \ll T$, which allows us to reduce the average per-iteration runtime. This lazy projection strategy, inspired by convex optimization techniques (see Section 4.4 in (Bubeck, 2015)), appears to be novel in a boosting context. We prove QuantumBoost's convergence in $T = O(\log(1/\epsilon)/\gamma^2)$ iterations by controlling the error (measured in terms of relative entropy) accumulated through the approximate projections and carefully choosing $K$.

3. As in the quantum algorithm presented in (Izdebski & de Wolf, 2023), examples for the weak learner may be prepared using amplitude amplification (Brassard et al., 2002). Even for the preparation of classical random examples, which are needed for a (classical) weak learner, first preparing a quantum example and then measuring it is more efficient than classical rejection-sampling.

While previous boosting methods either required the maintenance of an explicit $m$-dimensional weight vector, or computing an approximation of the sum of its elements, we only keep an implicit representation of the weight vector, which enables us to compute its entries with $O(t)$ runtime at iteration $t$ and to avoid the need of approximating its sum.

### 1.2. Comparison with related work

The runtime of boosting algorithms is often stated as a function of the Vapnik-Chervonenkis (VC) dimension $d$ of the hypothesis class associated with the weak learner (Vapnik & Chervonenkis, 1971). Boosting algorithms that combine $T$ weak hypotheses (one from each iteration) into one strong hypothesis, typically do this by taking the sign of a (possibly weighted) sum of the weak hypotheses. It is proven in (Shalev-Shwartz & Ben-David, 2014) that the VC-dimension of the hypothesis class of all such strong hypotheses is $\widetilde{O}(T \cdot d)$, and (by general VC-theory) a number of examples $m$ of that order then suffice for learning a good strong hypothesis. In order to contextualize our Theorem 1.1, we present the provable runtimes associated with all relevant boosting algorithms in Table 1, noting this correspondence between $m$ and $d$, and suppressing all polylog factors in the stated runtimes to improve readability.

Since there is no longer an explicit dependence on the number of examples $m$ in QuantumBoost, there is no subsequent explicit dependence on the VC-dimension $d$ either. This sounds too good to be true, and in some sense it is. A $\gamma$-weak learner produces a hypothesis that is promised to be correct on a $(1/2 + \gamma)$-fraction of the training set, with respect to any distribution and target function $f$ in a concept class $\mathcal{C}$. It can be shown that the VC-dimension $d$ of the weak learner's hypothesis class is at least roughly $\gamma^2$ times the VC-dimension $d_\mathcal{C}$ of the concept class $\mathcal{C}$ that the target function $f$ comes from. By general VC-theory again, the sample complexity of the weak learner (and hence its runtime $W$) must then be lower bounded by roughly $\gamma^2 d_\mathcal{C}$. Note that we cannot write upper bounds on runtime in terms of $d_C$ and $\gamma$ because we do not have an upper bound on $W$ in terms of $d_C$ and $\gamma$, only the mentioned lower bound (and the runtime $W$ of a learning algorithm can of course be *much* larger than its sample complexity). Boosting algorithms have to be able to work with any weak learner that can produce $(1/2 + \gamma)$-weak hypotheses, hence we leave $W$ as a placeholder here for the weak learner's runtime, and

also as an upper bound for its sample complexity.

We also note that the version of SmoothBoost by (Barak et al., 2009) could exploit implicit representations of weight vectors and avoid the explicit $d$-dependence as well. However, this would increase its $1/\gamma^6$ dependence to $1/\gamma^8$ for computing approximate Bregman projections with the implicit weight vectors. Our algorithm notably improves the scaling in $\epsilon$ and matches the best known scaling in $\gamma$, seen in AdaBoost.

## 2. Preliminaries

In this section we include some notation and helpful results. All $\log$s are natural logarithms, unless explicitly stated otherwise. Expressions like $\widetilde{O}(f(n))$ suppress polylogarithmic factors: $\widetilde{O}(f(n))$ is defined as $f(n)(\log n)^{O(1)}$.

### 2.1. Computational model

The computational model we assume here is a classical RAM model with a quantum co-processor. In addition to its classical operations, the classical machine can prepare a description of a quantum circuit and an initial computational basis state and send it to the quantum co-processor, which runs the circuit on the initial state, measures the final state, and returns the measurement outcome. We may fix any universal set of elementary quantum gates for our quantum circuits, for instance Hadamard, $T$, and CNOT-gates; the precise choice does not matter since each universal gate set can efficiently approximate the gates in any other gate set.

We assume the input bits, in particular the ones of the $m$ initial examples, are given in a quantum read-only classical memory (QCROM). The QCROM stores some $N$-bit string $z = z_0 \ldots z_{N-1}$, and we have a unitary $O_z$ available that maps $O_z : |i, b\rangle \mapsto |i, b \oplus z_i\rangle$ for all $i \in \{0, \ldots, N - 1\}$ and $b \in \{0, 1\}$. Such a unitary is called a "query (to $z$)". It can be used by the classical RAM machine, but can also be included in the description of the quantum circuits that the classical machine sends to the quantum co-processor; the quantum circuit may apply $O_z$ on superpositions. Like with classical ROM and RAM, we assume one QCROM query can be done very fast, at polylogarithmic cost in the memory-size $N$. We do not need any quantum-*writable* classical memory (QCRAM) in this paper. Besides the QCROM, whose contents do not change during the algorithm, we only need classical memory that is *not* accessed in superposition. It should be noted that QCROM (and even more so QCRAM) are controversial notions in quantum computing, since they are in practice very hard to implement fast in noisy quantum hardware. However, we feel that for a theory paper such as this, they are acceptable notions, since conceptually they just combine the (hopefully uncontroversial) notions of classical RAM and quantum

*Table 1.* Upper bounds on the runtimes of various boosting algorithms, suppressing polylogs.

| BOOSTING ALGORITHM | TOTAL RUNTIME | ITERATIONS ($T$) | REF. |
|---|---|---|---|
| 1. ADABOOST | $\frac{W}{\gamma^2} + \frac{d}{\epsilon\gamma^4}$ | $\frac{\log(1/\epsilon)}{\gamma^2}$ | (FREUND & SCHAPIRE, 1997) |
| 2. QUANTUM ADABOOST | $\frac{W^{1.5}\sqrt{d}}{\epsilon\gamma^{11}}$ | $\frac{\log(1/\epsilon)}{\gamma^2}$ | (ARUNACHALAM & MAITY, 2020) |
| 3. SMOOTHBOOST | $\frac{W}{\epsilon^2\gamma^2} + \frac{d}{\epsilon^4\gamma^4}$ | $\frac{1}{\epsilon\gamma^2}$ | (SERVEDIO, 2003) |
| 4. QUANTUM SMOOTHBOOST | $\frac{W}{\epsilon^{2.5}\gamma^4} + \frac{\sqrt{d}}{\epsilon^{3.5}\gamma^5}$ | $\frac{1}{\epsilon\gamma^2}$ | (IZDEBSKI & DE WOLF, 2023) |
| 5. KALE'S SMOOTHBOOST | $\frac{W}{\epsilon\gamma^2} + \frac{d}{\gamma^4} + \frac{1}{\epsilon\gamma^6}$ | $\frac{\log(1/\epsilon)}{\gamma^2}$ | (KALE, 2007; BARAK ET AL., 2009) |
| 6. QUANTUMBOOST | $\frac{W}{\sqrt{\epsilon}\gamma^4}$ | $\frac{\log(1/\epsilon)}{\gamma^2}$ | THIS WORK |

superposition.

When we refer to the cost or runtime of an algorithm or subroutine, we mean the total number of classical RAM operations used plus the total number of elementary gates in the quantum circuits sent to the quantum co-processor, counting a QCROM query as one gate (since our bounds will suppress polylogs, it doesn't really matter whether we treat the cost of a QCROM query as a constant or as polylog).

A boosting algorithm is a meta-algorithm to some extent: we can plug in an arbitrary weak learner $\mathcal{W}$ (quantum or classical) with its hypothesis class $\mathcal{H}$. We use $W$ to denote the runtime cost of our weak learner and also as a (possibly quite loose) upper bound on the number of (quantum or classical) examples that $\mathcal{W}$ uses. The hypothesis class $\mathcal{H}$ from which the weak hypotheses $h_t$ come, could also take many forms and we have to say something about how expensive it is to compute $h(x)$ for some $h \in \mathcal{H}$ and $x \in \mathcal{X}$. To abstract away from this, we will assume we have an oracle $O_{\mathcal{H}}$ available that maps $|h\rangle |x\rangle |b\rangle \mapsto |h\rangle |x\rangle |b \oplus h(x)\rangle$, i.e., that evaluates hypotheses $h$ for us at a given point $x$. We assume this $O_{\mathcal{H}}$ has $\widetilde{O}(1)$ cost, but this is merely a placeholder. For example, if $\mathcal{H}$ is the class of $n^2$-sized Boolean circuits, then the cost of running $O_{\mathcal{H}}$ would be $O(n^2)$, and all runtimes stated in the paper would have to be multiplied by this cost.

### 2.2. High-density measures and Bregman projections

We will need the following results about measures and projections in order to introduce the techniques behind boosting with smooth distributions.

**Definition 2.1** (High-density measures). Let $X$ be a finite set with discrete measure $M : X \to [0,1]$. Denote $|M| = \sum_{x \in X} M(x)$ as the weight of $M$ and $\mu(M) = |M|/|X|$ its density, which is a number in $[0,1]$. The set $\Gamma_\epsilon$ is the set of high-density measures, defined as

$$\Gamma_\epsilon = \{M \mid \mu(M) \geq \epsilon\}. \tag{6}$$

**Definition 2.2** (Smooth distributions). For a distribution $D$ on a finite set $X$, we say that $D$ is $\epsilon$-smooth if

$$||D||_\infty \leq \frac{1}{\epsilon \cdot |X|}, \tag{7}$$

where $||D||_\infty = \max_{x \in X} D(x)$. So no probability is more than a $1/\epsilon$-factor bigger than uniform. We denote the set of such distributions by $\mathcal{P}_\epsilon$.

*Fact* 2.3 (High-density and $\epsilon$-smoothness). Given a high-density measure $M : X \to [0,1]$ (i.e., $M \in \Gamma_\epsilon$), there is a natural induced probability distribution $D_M(x) = M(x)/|M|$ which is $\epsilon$-smooth, since $\frac{1}{|M|}M(x) \leq \frac{1}{\epsilon|X|}$ for all $x \in X$ if $\mu(M) \geq \epsilon$.

Fact 2.3 will come in handy when proving the performance guarantees of smooth boosting methods, since once we project onto the set of high-density measures, those measures, after normalization, are $\epsilon$-smooth distributions.

**Definition 2.4** (Bregman projection). The Bregman projection operator, which projects a measure $N$ onto the set of high-density measures $\Gamma_\epsilon$, is defined as follows

$$P_\epsilon(N) = \text{argmin}_{M \in \Gamma_\epsilon} \text{KL}(M||N), \tag{8}$$

where

$$\text{KL}(M||N) = \sum_x \left( M(x) \log \frac{M(x)}{N(x)} + N(x) - M(x) \right) \tag{9}$$

is the Kullback-Leibler (KL) divergence between measures $M$ and $N$. One can show that the "argmin" is unique, so $P_\epsilon$ is indeed a function.

**Theorem 2.5** (Bregman's theorem (Bregman, 1967)). *Let $M, M^*, N$ be measures such that $M \in \Gamma_\epsilon$ and $M^*$ is the exact projection of $N$ onto $\Gamma_\epsilon$. The following holds:*

$$\text{KL}(M||M^*) + \text{KL}(M^*||N) \leq \text{KL}(M||N).$$

Note that exactly computing the Bregman projection requires time linear in $|X|$ in general, to examine the weight of each $x \in X$. Luckily, Barak, Hardt, and Kale (Barak et al., 2009) proved that the output of the projection operator $P_\epsilon$ has an efficient implicit representation which will facilitate a much more efficient computation of an approximate projection.

**Lemma 2.6** (Implicit representation of Bregman projection). *Let $N$ be a measure with support at least $\epsilon|X|$ and $c \geq 1$ be the smallest constant such that the measure $M^* = \min(1, c \cdot N)$ has density $\epsilon$. Then $P_\epsilon(N) = M^*$.*

The proof is given in Lemma 3.1 of (Barak et al., 2009). Note that if we have a representation of $N$ available, then additionally storing the number $c$ gives us an (implicit) representation of $M^*$. In order to exploit this implicit representation, we need the notion of an *approximate* Bregman projection.

**Definition 2.7** (Approximate Bregman projection). Let $M^* = P_\epsilon(N)$ be the exact Bregman projection of $N$ onto $\Gamma_\epsilon$. Then $\widetilde{M}$ is an $\alpha$-approximation of $M^*$ if

1. $\widetilde{M} \in \Gamma_\epsilon$, and
2. $\mathrm{KL}(M||\widetilde{M}) \leq \mathrm{KL}(M||M^*) + \alpha \quad \forall\, M \in \Gamma_\epsilon$.

We also use the following fundamental identity, relating the KL-divergence between measures to the relative entropy (RE) between their normalized distributions.

*Fact* 2.8 (KL-RE Relation). The following identity holds for the KL-divergence between measures $A$ and $B$, and the relative entropy (RE) between their respective normalized distributions $D_A$ and $D_B$:

$$\mathrm{KL}(A||B) = |A|\,\mathrm{RE}(D_A||D_B) + |A|\log\left(\frac{|A|}{|B|}\right) + |B| - |A|,$$
(10)

where the relative entropy is defined as

$$\mathrm{RE}(D_A||D_B) = \sum_x D_A(x)\log\frac{D_A(x)}{D_B(x)}. \quad (11)$$

### 2.3. PAC ("Probably Approximately Correct") learning

For completeness, we explain weak and strong learners in the context of PAC learning, as well as the sample complexity bounds for generalization error guarantees in Appendix A.

### 2.4. Required quantum subroutines

QuantumBoost uses several quantum subroutines, which we detail in Appendix B. This includes amplitude estimation (Brassard et al., 2002), a particular state preparation technique (Izdebski & de Wolf, 2023) and quantum mean estimation (Brassard et al., 1998; Aaronson & Rall, 2020).

## 3. Smooth boosting

In Servedio's original SmoothBoost algorithm (Servedio, 2003), a smoothness condition is explicitly checked by summing $m$ components of the probability vector at every one of the $T$ iterations of the boosting procedure. Computing this sum, however, can be avoided completely by projecting onto the set of high-density measures, as discovered in (Kale, 2007; Barak et al., 2009). We present Kale's full algorithm in Algorithm 2 in Appendix C. Kale (Kale, 2007) proved that his SmoothBoost strategy outputs a hypothesis with low empirical error. The 01-loss vector (indexed by the $m$ data points), for a particular $h$, is defined as

$$\ell(x_i) = [h(x_i) = y_i], \quad (12)$$

where '$[\cdot]$' denotes truth value of a statement. Somewhat unintuitively, this loss is high (i.e., 1) when $x$ is correctly classified by $h$. This is due to the fact that the algorithm down-weights correctly-classified points, which only happens if they have high loss. The runtime costs incurred by Kale's approach involve: rejection sampling to generate an example for the weak learner at a runtime cost of $1/\epsilon$ to sample from $D^t$, which is done $W$ times; updating a weight vector using $O(m)$ runtime; and lastly, a projection which would incur a runtime of $O(m)$ if computed exactly.

Barak, Hardt and Kale (Barak et al., 2009) demonstrated that an *approximation* to this projection is still sufficient for correctness, and can be computed more efficiently by exploiting the implicit representation given in Lemma 2.6. We summarize their result in the following lemma.

**Lemma 3.1** (Computing the approximate Bregman projection (Barak et al., 2009)). *Let $N$ be a measure with exact projection onto the set $\Gamma_\epsilon$ given by $M^* = \min(1, c \cdot N)$ where $c \in [1, 1 + \rho]$ is unknown to the algorithm but $\rho$ is known. Suppose further that we can compute entries of the measure $N$ and sample uniform elements from the domain $X$ in runtime $t$. Then, an implicit representation (in the form of a constant $\tilde{c}$) of a $\zeta\epsilon|X|$-approximation of $M^*$ can be computed in runtime*

$$O\left(\frac{t}{\epsilon\zeta^2}(\log\log\frac{\rho}{\zeta} + \log\frac{1}{\delta})\log\frac{\rho}{\zeta}\right) \quad (13)$$

*with success probability $\geq 1 - \delta$.*

The proof in Lemma 3.2 of (Barak et al., 2009) uses binary search to find a constant $\tilde{c} \in [1, 1 + \rho]$ such that the measure $\widetilde{M} := \min(1, \tilde{c} \cdot N)$ satisfies

$$\epsilon \leq \mu(\widetilde{M}) \leq (1 + \zeta)\epsilon. \quad (14)$$

Having such a $\tilde{c}$ suffices to represent the desired approximate projection implicitly.

# 4. QuantumBoost

Now that we have stated all the necessary technical ingredients, we present QuantumBoost in Algorithm 1. QuantumBoost can be thought of as a quantized version of Kale's SmoothBoost algorithm (specifically the version with approximate Bregman projections (Barak et al., 2009)), coupled with a lazy projection strategy. Our algorithm performs the usual multiplicative updates at every iteration, but only enforces the high-density constraint (approximate projection onto $\Gamma_\epsilon$) once every $K$ iterations. In the subsequent analysis, we show that $T = O(\log(1/\epsilon)/\gamma^2)$ iterations still suffice.

---

**Algorithm 1** QuantumBoost Algorithm

---

**Require:** Parameters $\gamma \in (0, 1/2)$, $\epsilon \in (0, 1)$; Training set $S = \{(x_i, y_i)\}_{i=1}^m$; $\gamma$-weak quantum learner $\mathcal{W}$ with runtime $W$.

1: Initialize $M^1 \in \Gamma_\epsilon$ as the uniform measure with weight $|M^1| = \epsilon m$. Set the projection interval $K = 1/\gamma$ and the approximate-projection precision as $\zeta = \gamma/4$.

2: **for** $t = 1, \ldots, T$ **do**

3:    Using the state-preparation subroutine in Theorem B.2, prepare $W$ copies of the quantum state

$$\left| D^t \right\rangle = \sum_{x_i \in X} \sqrt{\frac{M^t(x_i)}{|M^t|}} \left| x_i, y_i \right\rangle, \qquad (15)$$

   corresponding to the normalized distribution $D^t = M^t/|M^t|$ and feed $|D^t\rangle^{\otimes W}$ to $\mathcal{W}$ to obtain hypothesis $h_t$ (we can compute the entries of the corresponding loss vector $\ell^t$ ourselves from this). Store (the name of) $h_t$ in classical memory.

4:    **if** $t \pmod K = 0$ **then**

5:       Compute $M^{t+1}$ as an implicit representation of the $\zeta \epsilon m$-approximate Bregman projection of $N^{t+1} = M^t(1 - \gamma)^{\ell^t}$ onto $\Gamma_\epsilon$ using the subroutine of Theorem 4.5. Store the corresponding constant $\tilde{c}_t$ in classical memory.

6:    **else**

7:       Set $M^{t+1} = N^{t+1} = M^t(1 - \gamma)^{\ell^t}$ (don't explicitly update an $m$-dimensional vector).

8:    **end if**

9: **end for**

10: **return** The final hypothesis $H(x) = \text{MAJ}(h_1(x), \ldots, h_T(x))$.

---

In contrast to classical boosters, QuantumBoost does not explicitly keep track of the $m$-dimensional weight vector $M^t$, but rather stores, after each iteration, the (name of the) weak hypothesis $h_t$ that was generated in that iteration, as well as the constant $\tilde{c}_t$ used to implicitly represent the approximate Bregman projection when such a projection is done. This information is stored in classical memory, and enables a quantum circuit to compute entries $M^t(x)$ on the fly with runtime $O(t)$. In each of the $T$ iterations, preparing copies of a quantum state for the weak learner, running the weak learner, and computing the approximate projection of an updated measure $N^{t+1} = M^t(1 - \gamma)^{\ell^t}$ at every $K^{\text{th}}$ iteration, are the main contributors to the overall runtime, which we analyze in Section 4.2.

## 4.1. Error bound for QuantumBoost

In order to prove that QuantumBoost outputs a hypothesis with low empirical error, we analyze the algorithm in two parts: the progress made by the usual multiplicative update rule and the accumulation of errors from the approximate projection at every $K^{\text{th}}$ iteration. To do this, we first state the approximate projection error in terms of relative entropy by using the KL-RE identity outlined in Fact 2.8.

**Lemma 4.1** (Approximate Bregman Projection and Relative Entropy). *Let $M_E$ be any measure with weight $|M_E| = \epsilon m$, and $D_E = M_E/|M_E|$ be its associated distribution. Let $M^{t+1}$ be a measure that is a $\zeta \epsilon m$-approximation of $N^{t+1} = M^t(1 - \gamma)^{\ell^t}$, $D^{t+1} = M^{t+1}/|M^{t+1}|$ the associated distribution of $M^{t+1}$ and $\hat{D}^{t+1} = N^{t+1}/|N^{t+1}|$ the associated distribution of $N^{t+1}$. Then,*

$$\text{KL}(M_E || M^{t+1}) - \text{KL}(M_E || M^*) \leq \zeta \epsilon m$$

*implies*

$$\text{RE}(D_E || D^{t+1}) - \text{RE}(D_E || D^*) \leq \zeta$$

*and*

$$\text{RE}(D_E || D^{t+1}) - \text{RE}(D_E || \hat{D}^{t+1}) \leq \zeta$$

*where $M^*$ is the exact projection of $N^{t+1}$, with associated distribution $D^* = M^*/|M^*|$.*

Essentially, Lemma 4.1 upper bounds the deviation in relative entropy (w.r.t. some reference measure $M_E$) between the distributions before and after the approximate projection. We defer the proof to Appendix D for better readability. With this lemma at hand, we can derive a general regret (loss) bound for QuantumBoost.

**Theorem 4.2** (Generalized Regret Bound for QuantumBoost). *Let QuantumBoost run for $T$ iterations with parameter $\gamma \in (0, 1/2)$, projection interval $K$, and precision $\zeta$. For every sequence of Boolean loss vectors $\ell^1, \ldots, \ell^T$, and every distribution $D \in \mathcal{P}_\epsilon$, the following regret bound holds:*

$$\sum_{t=1}^T \langle D^t, \ell^t \rangle \leq (1 + \gamma) \sum_{t=1}^T \langle D, \ell^t \rangle + \frac{R\zeta}{\gamma} + \frac{\text{RE}(D || D^1)}{\gamma}$$

*where $R = \lceil T/K \rceil$ is the total number of approximate Bregman projections that the algorithm makes.*

*Proof.* Fix a distribution $D \in \mathcal{P}_\epsilon$. We consider the potential function $\Psi^t(D) = \mathrm{RE}(D||D^t)$, and analyze its change $\Delta\Psi^t(D) = \Psi^{t+1}(D) - \Psi^t(D)$ in every iteration. We decompose the change into the update phase and the projection phase, using the intermediate distribution $\hat{D}^{t+1}$ (which corresponds to $N^{t+1}$ normalized, so after the update but before the projection):

$$\Delta\Psi^t(D) = \underbrace{\left[ \mathrm{RE}(D||\hat{D}^{t+1}) - \mathrm{RE}(D||D^t) \right]}_{\Delta\Psi^t_{Update}(D)}$$
$$+ \underbrace{\left[ \mathrm{RE}(D||D^{t+1}) - \mathrm{RE}(D||\hat{D}^{t+1}) \right]}_{\Delta\Psi^t_{Proj}(D)}.$$

We separately upper bound the two terms on the right-hand side, starting with $\Delta\Psi^t_{Update}(D)$. The transition from $D^t$ to $\hat{D}^{t+1}$ follows the multiplicative update rule: $\hat{D}^{t+1}(x) = \frac{1}{Z_t}D^t(x)(1-\gamma)^{\ell^t(x)}$ where $Z_t = \sum_x D^t(x)(1-\gamma)^{\ell^t(x)} = \sum_x D^t(x)(1-\gamma\ell^t(x)) = 1 - \gamma\langle D^t, \ell^t \rangle$ is the normalization factor (we used the fact that the entries of the loss vector are Boolean, so $(1-\gamma)^{\ell^t(x)} = 1 - \gamma\ell^t(x)$). Since

$$\mathrm{RE}(D||\hat{D}^{t+1}) - \mathrm{RE}(D||D^t) = \sum_x D(x) \log \frac{D^t(x)}{\hat{D}^{t+1}(x)}$$
$$= \sum_x D(x) \log \frac{Z_t}{(1-\gamma)^{\ell^t(x)}},$$

we have

$$\Delta\Psi^t_{Update}(D) = \log(Z_t) - \langle D, \ell^t \rangle \log(1-\gamma).$$

Using the inequalities $\log(Z_t) \leq -\gamma\langle D^t, \ell^t \rangle$ and $-\log(1-\gamma) \leq \gamma(1+\gamma)$ (valid for $\gamma \leq 1/2$):

$$\Delta\Psi^t_{Update}(D) \leq \gamma\left( \langle D, \ell^t \rangle(1+\gamma) - \langle D^t, \ell^t \rangle \right) \quad (16)$$

Next, we bound the approximate projection error $\Delta\Psi^t_{Proj}(D)$. If no projection occurs (which is actually the case in most iterations), then $D^{t+1} = \hat{D}^{t+1}$ and hence $\Delta\Psi^t_{Proj}(D) = 0$. If a projection occurs, we may bound the increase in RE potential using Lemma 4.1. Note that for an arbitrary $D \in \mathcal{P}_\epsilon$, we can define a corresponding measure $M_D(x) = D(x)\epsilon m$. Since $\|D\|_\infty \leq 1/(\epsilon m)$, we have $M_D(x) \leq 1$. The total weight is $|M_D| = \sum_x D(x)\epsilon m = \epsilon m$. Thus, $M_D \in \Gamma_\epsilon$ and Lemma 4.1 applies:

$$\Delta\Psi^t_{Proj}(D) \leq \zeta. \quad (17)$$

We now sum the (upper bounds on the) changes in the potential over all $T$ iterations:

$$\Psi^{T+1}(D) - \Psi^1(D) = \sum_{t=1}^T \Delta\Psi^t_{Update}(D) + \sum_{t=1}^T \Delta\Psi^t_{Proj}(D)$$
$$\leq \sum_{t=1}^T \gamma\left( \langle D, \ell^t \rangle(1+\gamma) - \langle D^t, \ell^t \rangle \right) + R\zeta.$$

Rearranging the terms to isolate the algorithm's overall loss $\sum_t \langle D^t, \ell^t \rangle$, and using the fact that relative entropy is non-negative ($\Psi^{T+1}(D) \geq 0$) and

$$\gamma\sum_{t=1}^T \langle D^t, \ell^t \rangle \leq \gamma(1+\gamma)\sum_{t=1}^T \langle D, \ell^t \rangle + R\zeta$$
$$+ \Psi^1(D) - \Psi^{T+1}(D)$$
$$\leq \gamma(1+\gamma)\sum_{t=1}^T \langle D, \ell^t \rangle + R\zeta + \mathrm{RE}(D||D^1).$$

Dividing by $\gamma$ proves the theorem. $\qquad\square$

Using the regret bound, we now prove the error guarantees achieved by QuantumBoost.

**Theorem 4.3** (Empirical error bound for QuantumBoost). *Given access to a $\gamma$-weak learner for the concept class $\mathcal{C}$ with hypothesis class $\mathcal{H}$ and a training set $S = \{(x_i, y_i)\}_{i=1}^m$, QuantumBoost outputs a hypothesis $H$ with empirical error*

$$\widehat{\mathrm{err}}(H) = \Pr_{i\sim[m]}[H(x_i) \neq y_i] < \epsilon \quad (18)$$

*It uses $T = O(\log(1/\epsilon)/\gamma^2)$ iterations, with one call to the weak learner per iteration.*

*Proof.* We use the same potential function $\Psi^t(D) = \mathrm{RE}(D||D^t)$ as the previous proof. We want to upper bound the size of the set $E$ of instances $x_i$ that are misclassified by the final hypothesis $H$. Suppose, towards a contradiction, that $|E| \geq \epsilon m$. Let $D_E$ be the uniform distribution over the set $E$.

We analyze the total change in the potential over all $T$ iterations by a telescoping sum:

$$\Delta\Psi_{Total}(D) = \Psi^{T+1}(D) - \Psi^1(D) \quad (19)$$
$$= \sum_{t=1}^T (\Delta\Psi^t_{Update}(D) + \Delta\Psi^t_{Proj}(D))$$
$$\quad (20)$$

which holds for any $D$. Fixing $D$ as $D_E$, we use the previously derived bounds of Equation (16) and (17). Summing over $T$ iterations gives

$$\sum_{t=1}^T \Delta\Psi^t_{Update}(D_E)$$
$$\leq \sum_{t=1}^T \gamma\left( \langle D_E, \ell^t \rangle(1+\gamma) - \langle D^t, \ell^t \rangle \right)$$
$$= \gamma\left( (1+\gamma)\sum_{t=1}^T \langle D_E, \ell^t \rangle - \sum_{t=1}^T \langle D^t, \ell^t \rangle \right)$$
$$\leq \gamma((1+\gamma)T/2 - (1/2+\gamma)T)) = -T\gamma^2/2$$

where the second inequality uses $\sum_{t=1}^{T}\langle D_E, \ell^t\rangle \leq T/2$ because $E$ is defined as the set of instances of $X$ where the final hypothesis $H$ (i.e., the majority vote of all $h_t$) fails; and it uses $\langle D^t, \ell^t\rangle \geq 1/2 + \gamma$ which comes from the definition of our weak learner. Furthermore, the aggregate error from the $R$ approximate projections is $\sum_{t=1}^{T} \Delta\Psi_{Proj}^t(D_E) \leq R\zeta$.

QuantumBoost sets the projection interval to $K = 1/\gamma$, so the total number of (approximate) projections is $R = T/K = T\gamma$ (assuming for simplicity that $1/\gamma$ and $T/K$ are integers). It set the precision parameter $\zeta = \gamma/4$. Hence the aggregate projection error is at most:

$$R\zeta = (T\gamma)\left(\frac{\gamma}{4}\right) = \frac{T\gamma^2}{4}.$$

The total change in potential is therefore upper bounded by something negative:

$$\Delta\Psi_{Total}(D_E) \leq -\frac{T\gamma^2}{2} + \frac{T\gamma^2}{4} = -\frac{T\gamma^2}{4}. \qquad (21)$$

Because $D^1$ is the uniform distribution over the training set $X$, i.e. $D^1(x) = 1/m$, we have

$$\Psi^1(D_E) = \mathrm{RE}(D_E||D^1) = \sum_{x \in E} D_E(x) \log\left(\frac{D_E(x)}{D^1(x)}\right).$$

Since $D_E(x) = 1/|E|$ for $x \in E$,

$$\Psi^1(D_E) = \sum_{x \in E} \frac{1}{|E|} \log\left(\frac{1/|E|}{1/m}\right) = \log\left(\frac{m}{|E|}\right)$$
$$\leq \log(1/\epsilon).$$

Using the fact that $\Psi^{T+1}(D_E) \geq 0$, we get $\Delta\Psi_{Total}(D_E) = \Psi^{T+1}(D_E) - \Psi^1(D_E) \geq -\log(1/\epsilon)$. Equation (21) implies

$$\frac{T\gamma^2}{4} \leq \log(1/\epsilon).$$

Accordingly, if $T > \frac{4\log(1/\epsilon)}{\gamma^2}$, then we obtain a contradiction, so there cannot exist a set $E$ of incorrectly classified instances of size $|E| \geq \epsilon m$ in the training set. Hence, with $T = \lfloor\frac{4\log(1/\epsilon)}{\gamma^2}\rfloor + 1$, QuantumBoost achieves empirical error $\Pr_{i\sim[m]}[H(x_i) \neq y_i] < \epsilon$. $\qquad \square$

By applying Theorem 4.3 with an empirical-error bound of $\epsilon/2$, combined with Claim A.3 with $\eta = \epsilon/2$, we may also bound the *generalization* error performance of Quantum-Boost, assuming a sufficiently large training set.

**Corollary 4.4** (Generalization error of QuantumBoost). *Let $d$ be the VC-dimension of the weak learner's hypothesis class $\mathcal{H}$. Assume the number of examples is at least*

$$m = \Theta\left(\frac{d\log(d/\epsilon)}{\gamma^2\epsilon^2} + \frac{\log(1/\delta)}{\epsilon^2}\right),$$

*with a sufficiently large constant in the $\Theta(\cdot)$. Then, for every $f \in \mathcal{C}$ and every distribution $D$ on the domain of $f$, QuantumBoost outputs a hypothesis $H$ that, with probability $\geq 1 - \delta$ (probability taken over the choice of the training set and the internal randomness of the algorithm), has generalization error*

$$\mathrm{err}(H) = \Pr_{x\sim D}[H(x) \neq f(x)] \leq \epsilon.$$

### 4.2. Overall runtime of QuantumBoost

We use quantum techniques to improve the runtime for computing approximate Bregman projections.

**Theorem 4.5** (Faster approximate Bregman projection). *Let $N$ be a measure on domain $X$ with exact projection onto the set $\Gamma_\epsilon$ given by $M^* = \min(1, c \cdot N)$ where $1 \leq c \leq 1 + \rho$. Suppose further that we can compute entries of the measure $N$ and generate uniform superpositions over the domain $X$ in runtime $t$. Then, an implicit representation of a $\zeta\epsilon|X|$-approximation of $M^*$ (in the form of a constant $\tilde{c}$) can be computed on a quantum computer in runtime*

$$O\left(\frac{t}{\sqrt{\epsilon}\zeta}(\log\log\frac{\rho}{\zeta} + \log(1/\delta))\log\frac{\rho}{\zeta}\right) = \widetilde{O}\left(\frac{t}{\sqrt{\epsilon}\zeta}\right) \tag{22}$$

*with success probability $\geq 1 - \delta$.*

*Proof.* By Lemma 3.1 of (Barak et al., 2009), it suffices to find a constant $\tilde{c} \in [1, 1 + \rho]$ such that the measure $\widetilde{M} := \min(1, \tilde{c} \cdot N)$ satisfies $\epsilon \leq \mu(\widetilde{M}) \leq (1 + \zeta)\epsilon$. Using binary search, in combination with the faster mean estimation of Theorem B.3, we may find such a constant in the stated runtime using binary search. Binary search has $\log\left(\frac{\rho}{\zeta}\right)$ steps, each of which involves a single call to the algorithm of Theorem B.3 with runtime $\widetilde{O}(t/(\sqrt{\epsilon}\zeta))$ and error probability set to $\delta' \leq \delta/\log(\rho/\zeta)$. Taking a union bound over the number of binary search steps shows that they all succeed except with error probability $\leq \delta'\log(\rho/\zeta) \leq \delta$. $\qquad \square$

Lastly, we upper bound the runtime of QuantumBoost.

**Theorem 4.6** (Runtime of QuantumBoost). *Given access to a $\gamma$-weak learner with runtime $W$ for the concept class $\mathcal{C}$ with hypothesis class $\mathcal{H}$, and a training set $S = \{(x_i, y_i)\}_{i=1}^{m}$, QuantumBoost (Algorithm 1) produces (with probability $\geq 1 - \delta$) a hypothesis with generalization error at most $\epsilon$, using $m = \widetilde{\Theta}(d/(\gamma^2\epsilon^2))$ examples where $d$ is the VC-dimension of $\mathcal{H}$. QuantumBoost makes $T = O(\log(1/\epsilon)/\gamma^2)$ calls to the weak learner and has overall runtime*

$$\widetilde{O}\left(\frac{W}{\sqrt{\epsilon}\gamma^4}\right).$$

*Proof.* Due to the recursive structure of $M^t$, keeping track of the constants $\tilde{c}_t$ at every $K^{\text{th}}$ iteration, along with the hypotheses $h_t$, enables us to compute entries of $M^t$ in $O(t)$ runtime. At iteration $t$, the weak learner takes $O(W)$ quantum examples as an input, defined as

$$|D^t\rangle = \sum_{x_i \in X} \sqrt{\frac{M^t(x_i)}{|M^t|}} \, |x_i, y_i\rangle. \qquad (23)$$

where $X := \{x_i : (x_i, y_i) \in S\}$. Since the weight $|M^t|$ decreases by at most a factor $(1 - \gamma)$ per iteration, over a sequence of $K = 1/\gamma$ (non-projecting) iterations, the decrease is lower bounded by a factor $(1 - \gamma)^K \approx e^{-\gamma K} = \Omega(1)$. After every $K$ iterations, we (approximately) project the measure back onto the set $\Gamma_\epsilon$ of measures of weight $\geq \epsilon m$. These two things together ensure that $|M^t| = \Omega(\epsilon m)$ throughout the algorithm, so we can prepare a quantum example with $\widetilde{O}(1/\sqrt{\epsilon})$ operations, times the runtime incurred for computing entries of the $M^t$ vector, using the state-preparation subroutine in Theorem B.2. The aggregate runtime incurred for example-preparation over all $T = \widetilde{O}(1/\gamma^2)$ calls to the weak learner, is then

$$\widetilde{O}\left(T \cdot W \cdot \frac{T}{\sqrt{\epsilon}}\right) = \widetilde{O}\left(\frac{W}{\sqrt{\epsilon}\gamma^4}\right). \qquad (24)$$

The approximate Bregman projection subroutine at iteration $t$ has runtime $\widetilde{O}(t/(\sqrt{\epsilon}\zeta))$ by Theorem 4.5. By Theorem 4.3, setting $\zeta = \gamma/4$ and summing over the $R = T/K = 1/\gamma$ iterations where projections occur, yields an aggregate runtime for the projections of

$$\widetilde{O}\left(\frac{R \cdot T}{\sqrt{\epsilon}\zeta}\right) = \widetilde{O}\left(\frac{(1/\gamma)(1/\gamma^2)}{\sqrt{\epsilon}\gamma}\right) = \widetilde{O}\left(\frac{1}{\sqrt{\epsilon}\gamma^4}\right). \quad (25)$$

The total runtime for QuantumBoost is the sum of (24) and (25), concluding the proof. $\qquad\square$

### 4.3. Further remarks on boosting

It is worth noting that the optimal sample complexity for general boosting algorithms is $\tilde{\Theta}(d/(\gamma^2\epsilon))$ (Green Larsen & Ritzert, 2022). The sample complexity of QuantumBoost is $\tilde{O}(d/(\gamma^2\epsilon^2))$, which has an extra $1/\epsilon$ factor. However, this matches the best known sample complexity $O(d/(\gamma^2\epsilon^2))$ among all known *smooth* boosting algorithms, and is optimal up to logarithmic factors given the corresponding lower bound $\tilde{\Omega}(d/(\gamma^2\epsilon^2))$ in (Blanc et al., 2024). Our main contribution lies in the improved runtime (the amount of work done around each call to the weak learner is only polylogaritmic in $m$, not polynomial like all earlier classical and quantum boosting algorithms), not the sample complexity.

In order to reduce the number of symbols in our presentation, we chose in this paper to use '$W$' both as an upper bound for the runtime of the weak learner, and for the number of

examples it needs. We could complicate notation by having a separate symbol, say $w$, for the sample complexity of the weak learner, and continue to use $W$ for its runtime. Potentially $w \ll W$, in which case our earlier accounting overestimated the cost of preparing the examples for the calls to the weak learner. QuantumBoost's runtime upper bound would then be $\dfrac{W}{\gamma^2} + \dfrac{w}{\sqrt{\epsilon}\gamma^4}$ and AdaBoost's would be $\dfrac{W}{\gamma^2} + \dfrac{m + w}{\gamma^2}$ (up to log-factors).

Both QuantumBoost and AdaBoost use roughly $1/\gamma^2$ iterations, each of which invokes the weak learner once. The classical lower bound of $\Omega(1/\gamma^2)$ on the number of calls to the weak learner (and thus, the number of iterations) also holds for quantum boosters like ours by adapting the constructions in (Karbasi & Larsen, 2024; Lyu et al., 2024), so the $W/\gamma^2$ dependence is best-possible. Also, both QuantumBoost and AdaBoost have overall runtime linear in $1/\gamma^4$. In the case of AdaBoost, this is because in each of its $1/\gamma^2$ iterations, AdaBoost updates each of the weights of the $m$ examples, and $m$ is linear in $1/\gamma^2$. In the case of QuantumBoost, this is because in each of the $1/\gamma^2$ iterations, QuantumBoost generates $w$ examples for the weak learner, and each example-generation has a cost that is linear in $T$, and hence, linear in $1/\gamma^2$. The $\epsilon$-dependence of QuantumBoost is significantly better than AdaBoost's because the latter's sample size $m$ is at least $d/\epsilon\gamma^2$, where $d$ is the VC-dimension of the weak learner's hypothesis class.

## 5. Future work

We mention a few questions for future work:

- Can we improve the 4th-power dependence on $\gamma$ of QuantumBoost to something better?

- As raised in (Izdebski & de Wolf, 2023), can QuantumBoost be modified for an *agnostic* learning setting, where $(x, y)$ pairs follow a joint distribution on $\mathcal{X} \times \mathcal{Y}$?

- Lastly, are there practical applications where QuantumBoost outperforms (at least in theory) all known boosting algorithms?

## Acknowledgements

We acknowledge Deep Think, an LLM produced by Google DeepMind, which was instrumental in proving error convergence using lazy projections. While the core idea for using lazy projections was generated by the authors, Deep Think recognized that the proof technique using KL-divergence (which is what we originally tried) would not work, and switching to relative entropy allowed us to prove the result. We further thank Jarrod McClean, Robin Kothari, Dar Gilboa and Sid Jain for useful discussions about the paper.

## Impact Statement

This paper presents a faster algorithm with a rigorous mathematical analysis for boosting a weak learner to a strong learner on a *quantum* computer. We do not anticipate strong societal consequences of our work, at least not until a large-scale quantum computer becomes available. We do acknowledge the possibility of a large resource footprint and energy cost required to maintain coherent QCROM queries over large datasets, but this is still rather unclear and further research progress needs to be made to clarify such a point.

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

# A. PAC learning basics

**Definition A.1** ($(\epsilon, \delta)$-PAC learner (Valiant, 1984)). An algorithm $\mathcal{A}$ is an $(\epsilon, \delta)$-PAC learner for concept class $\mathcal{C}$ with hypothesis class $\mathcal{H}$ if, for every target function $f \in \mathcal{C}$ and every distribution $D$ on $f$'s domain $\mathcal{X}$, $\mathcal{A}$ outputs a hypothesis $h \in \mathcal{H}$ with small generalization error:

$$\text{err}(h) = \Pr_{x \sim D}[h(x) \neq f(x)] \leq \epsilon \tag{26}$$

with success probability $1 - \delta$ over randomly drawn examples $\{(x_i, f(x_i))\}_{i=1}^{m}$, where the $x_i$'s are drawn i.i.d. from $D$.

This setting is sometimes called *distribution-independent* learning, since the same learning algorithm $\mathcal{A}$ should work for any distribution $D$. The number $\epsilon$ is typically referred to as the *generalization error* of the hypothesis (or of $\mathcal{A}$). The number of examples that are necessary and sufficient for learning depends on the VC-dimension (Vapnik & Chervonenkis, 1974) of the relevant concept class and the desired generalization error. The VC-dimension is defined as follows: a set $W \subseteq \mathcal{X}$ is *shattered* by $\mathcal{H}$ if, for each of the $2^{|W|}$ possible binary labelings of the elements of $W$, there is an $h \in \mathcal{H}$ consistent with that labeling; then VC-dim($\mathcal{H}$) is the size $|W|$ of a largest shattered set $W$.

A $\gamma$-weak learner is simply a $(1/2 - \gamma, 0)$-PAC learner for the concept class $\mathcal{C}$ with a hypothesis that comes from $\mathcal{H}$, given access to a training set. Boosting combines hypotheses generated by a weak learner to form a new hypothesis $H$. Since $H$ is a combination of several weak hypotheses $h \in \mathcal{H}$, $H$ subsequently belongs to a concept class much larger than $\mathcal{H}$. Letting $\bar{\mathcal{H}}$ denote this larger class to which $H$ belongs, we may think of a strong learner as an $(\epsilon, \delta)$-PAC learner with hypothesis class $\bar{\mathcal{H}}$. The following result illustrates the relationship between the VC-dimensions of $\mathcal{H}$ and $\bar{\mathcal{H}}$ in the case where the weak hypotheses are combined by a majority vote.

*Claim* A.2 (VC-dimension of $\bar{\mathcal{H}}$). Let $d$ denote the VC-dimension of the concept class $\mathcal{H}$. Then the hypothesis class $\bar{\mathcal{H}} = \{\text{MAJ}(h_1, \ldots, h_T) \mid h_i \in \mathcal{H}\}$ has VC-dimension $\widetilde{O}(T \cdot d)$.

The proof can be found in pg. 109 of (Shalev-Shwartz & Ben-David, 2014).

**Classical and quantum examples:** Classically, one provides learners with $m$ examples, which are random variables of the form $(x_i, y_i)$. Quantum learners, however, are given access to $m$ copies of the state

$$\sum_{x_i \in X} \sqrt{D(x_i)} |x_i, y_i\rangle. \tag{27}$$

One can think of this "quantum example" state as a coherent version of the classical random example. The quantum learner can perform a POVM measurement over the copies, where each outcome is associated with a hypothesis. Even with this ability, it turns out that the number of classical and quantum examples needed for PAC learning are the same up to a constant factor; in other words, having quantum examples available does not significantly reduce the sample complexity in distribution-independent PAC learning (Arunachalam & de Wolf, 2018). For our purposes, in boosting, we are initially given $m$ classical examples, but we actually allow weak *quantum* learners, thus making the class of weak learners that we can boost more general and more powerful. Accordingly, our boosting algorithm will have to prepare the quantum examples that are fed to a weak quantum learner at each iteration. We explicitly account for this example-preparation cost in QuantumBoost.

**Empirical error vs generalization error:** Another PAC learning result that will be important for the error guarantee of QuantumBoost, relates the generalization error of a hypothesis to its empirical error. The empirical error of a hypothesis $h \in \mathcal{H}$ w.r.t. a training set $S = \{(x_i, y_i)\}_{i=1}^{m}$ is

$$\widehat{\text{err}}(h) = \Pr_{i \sim [m]}[h(x_i) \neq y_i] \tag{28}$$

where $i \sim [m]$ denotes that $i$ is taken uniformly at random from $[m] = \{1, 2, \ldots, m\}$. The empirical error of $h$ only depends on the training set $S$. In contrast, its generalization error $\text{err}(h) = \Pr_{x \sim D}[h(x) \neq f(x)]$ depends on both the distribution $D$ and the target function $f$. The following claim implies that if $m$ is large enough, then small empirical error implies small generalization error.

*Claim* A.3. [Generalization error and empirical error] Let $d$ be the finite VC-dimension of the hypothesis class $\mathcal{H}$. For a randomly chosen training set $S$ of size $m$ and any $\eta > 0$,

$$\Pr[\exists\, h \in \mathcal{H} : \text{err}(h) - \widehat{\text{err}}(h) > \eta] \leq 8 \left(\frac{em}{d}\right)^d \exp\left(-\frac{m\eta^2}{32}\right),$$

where the generalization error $\mathrm{err}(h)$ is taken with respect to the target function $f$ from concept class $\mathcal{C}$ and the empirical error $\widehat{\mathrm{err}}(h)$ is with respect to the training set $S$.

The proof can be found in Theorem 2.5 of (Schapire & Freund, 2013). We make use of this claim in Section 4.1 (Corollary 4.4) when bounding the generalization error achieved by QuantumBoost.

## B. Quantum subroutines

For better readability, we deferred the following subroutines to this appendix.

**Theorem B.1** (Amplitude estimation (Brassard et al., 2002)). *Let $\delta \in (0,1)$. Given a natural number $A$ and access to an $(n+1)$-qubit unitary $U$ satisfying*

$$U |0^n\rangle |0\rangle = \sqrt{a} |\phi_0\rangle |0\rangle + \sqrt{1-a} |\phi_1\rangle |1\rangle, \tag{29}$$

*where $|\phi_0\rangle$ and $|\phi_1\rangle$ are arbitrary $n$-qubit states and $a \in [0,1]$, there exists a quantum algorithm that uses $O(A\log(1/\delta))$ applications of $U$ and $U^\dagger$ and $\widetilde{O}(A\log(1/\delta))$ elementary gates, and outputs an estimator $\lambda$ such that, with probability $\geq 1 - \delta$,*

$$|\lambda - \sqrt{a}| \leq \frac{1}{A}. \tag{30}$$

**Theorem B.2** (State preparation (Izdebski & de Wolf, 2023)). *Assume query access to the numbers $M_1, M_2, \ldots, M_m \in [0,1]$ with an unknown sum $\sum_{i=1}^m M_i$ which has a known lower bound of $\epsilon m$. Then we can prepare the following quantum state*

$$\sum_{i=1}^m \sqrt{\frac{M_i}{|M|}} |i\rangle, \tag{31}$$

*using an expected number of $O(1/\sqrt{\epsilon})$ queries and $\widetilde{O}(1/\sqrt{\epsilon})$ other operations.*

This technique allows us to prepare a quantum example for an $\epsilon$-smooth distribution $D = M/|M|$ on the training set, assuming we can query the entries of the measure $M$.

The next well-known theorem is a key ingredient to speeding up the computation of approximate Bregman projections, shown in Section 4.2.

**Theorem B.3** (Mean estimation (Brassard et al., 1998; Aaronson & Rall, 2020)). *Let $\epsilon, \delta \in (0,1)$ and $\zeta \in (0, 0.5)$. Furthermore, let $x_1, \ldots, x_N \in [0,1]$ with (unknown) mean $\mu = \frac{1}{N} \sum_{i \in [N]} x_i$. Suppose we have quantum query access to $O_x : |i\rangle |0\rangle \to |i\rangle |x_i\rangle$ and we know that $\mu \geq \epsilon/2$. Then there exists a quantum algorithm that, with success probability $\geq 1 - \delta$, estimates $\mu$ with multiplicative error $\zeta$ using $\widetilde{O}\left(\frac{\log(1/\delta)}{\sqrt{\epsilon}\zeta}\right)$ queries to $O_x$ and its inverse, and similarly, $\widetilde{O}\left(\frac{\log(1/\delta)}{\sqrt{\epsilon}\zeta}\right)$ other operations.*

*Proof.* We need to output an estimator $\hat{\mu}$, such that $|\hat{\mu} - \mu| \leq \mu\zeta$ with success probability $1 - \delta$. We first show that we can implement $U_\mu : |0\rangle |0\rangle \to \sqrt{\mu} |\phi_1\rangle |1\rangle + \sqrt{1-\mu} |\phi_0\rangle |0\rangle$ (for some normalized states $|\phi_0\rangle, |\phi_1\rangle$) using one query to $O_x$ and $\widetilde{O}(1)$ other elementary gates. Define the controlled rotation unitary as follows, such that for each $a \in [0,1]$

$$U_{CR} : |a\rangle |0\rangle \to |a\rangle (\sqrt{a} |1\rangle + \sqrt{1-a} |0\rangle).$$

This can be implemented up to negligibly small error by $\widetilde{O}(1)$ elementary gates. Starting with the easy-to-prepare state $\frac{1}{\sqrt{N}} \sum_{i \in [N]} |i\rangle |0\rangle |0\rangle$, apply $O_x$ on the first two registers, followed by $U_{CR}$ on the second and third registers to obtain

$$\frac{1}{\sqrt{N}} \sum_{i \in [N]} |i\rangle |x_i\rangle \left(\sqrt{x_i} |1\rangle + \sqrt{1-x_i} |0\rangle\right) \equiv \sqrt{\mu} |\phi_1\rangle |1\rangle + \sqrt{1-\mu} |\phi_0\rangle |0\rangle.$$

By Theorem B.1, we can find an estimator $\lambda$ such that with probability $\geq 1 - \delta$,

$$|\lambda - \sqrt{\mu}| \leq \sqrt{\epsilon}\zeta/2,$$

using time $\widetilde{O}\left(\frac{\log(1/\delta)}{\sqrt{\epsilon}\zeta}\right)$ and applications to $U_\mu$ and its inverse. Hence we can output an estimator $\hat{\mu} = \lambda^2$ satisfying

$$
\begin{aligned}
|\hat{\mu} - \mu| &= |\lambda^2 - \mu| \\
&= |\lambda + \sqrt{\mu}| \cdot |\lambda - \sqrt{\mu}| \\
&\leq (2\sqrt{\mu} + \sqrt{\epsilon}\zeta/2) \cdot \sqrt{\epsilon}\zeta/2 \\
&\leq \mu\zeta,
\end{aligned}
$$

where the last inequality used $\epsilon/2 \leq \mu$ and $\zeta \leq 1/2$. We used $\widetilde{O}\left(\frac{\log(1/\delta)}{\sqrt{\epsilon}\zeta}\right)$ time and applications of $U_\mu$ and its inverse. Since we can implement $U_\mu$ using just one query to $O_x$, we have finished the proof. $\qquad\square$

## C. Kale's Smoothboost Algorithm

We present Kale's version of smoothboost in Algorithm 2.

---
**Algorithm 2** Kale's SmoothBoost Algorithm

---
**Require:** Parameters $\gamma \in (0, 1/2)$ and $\epsilon \in [0, 1]$; training set $S = \{(x_i, y_i)\}_{i=1}^m$; a $\gamma$-weak learner $\mathcal{W}$ with runtime $W$.
 1: Initialize $M^1 \in \Gamma_\epsilon$ as the uniform measure with weight $|M^1| = \epsilon m$.
 2: **for** $t = 1, \ldots, T$ **do**
 3:     Feed $W$ examples generated according to the distribution $D^t = M^t/|M^t|$ to $\mathcal{W}$ to obtain $h_t$.
 4:     Observe the associated loss vector $\ell^t \in \{0, 1\}^m$.
 5:     Compute $M^{t+1}$ as the projection of $N^{t+1} = M^t(1 - \gamma)^{\ell^t}$ onto the set $\Gamma_\epsilon$.
 6: **end for**
 7: **return** The final hypothesis $H(x) = \text{MAJ}(h_1(x), \ldots, h_T(x))$.

---

## D. Approximate Bregman projections in relative entropy terms

**Lemma 4.1** (Approximate Bregman Projection and Relative Entropy). *Let $M_E$ be any measure with weight $|M_E| = \epsilon m$, and $D_E = M_E/|M_E|$ be its associated distribution. Let $M^{t+1}$ be a measure that is a $\zeta\epsilon m$-approximation of $N^{t+1} = M^t(1 - \gamma)^{\ell^t}$, $D^{t+1} = M^{t+1}/|M^{t+1}|$ the associated distribution of $M^{t+1}$ and $\hat{D}^{t+1} = N^{t+1}/|N^{t+1}|$ the associated distribution of $N^{t+1}$. Then,*

$$\text{KL}(M_E||M^{t+1}) - \text{KL}(M_E||M^*) \leq \zeta\epsilon m$$

*implies*

$$\text{RE}(D_E||D^{t+1}) - \text{RE}(D_E||D^*) \leq \zeta$$

*and*

$$\text{RE}(D_E||D^{t+1}) - \text{RE}(D_E||\hat{D}^{t+1}) \leq \zeta$$

*where $M^*$ is the exact projection of $N^{t+1}$, with associated distribution $D^* = M^*/|M^*|$.*

*Proof.* Assume

$$\text{KL}(M_E||M^{t+1}) \leq \text{KL}(M_E||M^*) + \zeta\epsilon m. \tag{32}$$

Expanding both sides using the KL-RE identity from Fact 2.8 and the fact that $M^{t+1}$ is a $\zeta\epsilon m$-approximate measure of $M^*$, we obtain the following.

**LHS Expansion**

$$
\begin{aligned}
\text{KL}(M_E||M^{t+1}) &= |M_E| \, \text{RE}(D_E||D^{t+1}) + |M_E| \log\left(\frac{|M_E|}{|M^{t+1}|}\right) + |M^{t+1}| - |M_E| \\
&= \epsilon m \, \text{RE}(D_E||D^{t+1}) + \epsilon m \log\left(\frac{\epsilon m}{(1+\zeta)\epsilon m}\right) + ((1+\zeta)\epsilon m - \epsilon m) \\
&= \epsilon m \left[\text{RE}(D_E||D^{t+1}) - \log(1 + \zeta) + \zeta\right]
\end{aligned}
$$

**RHS Expansion**   Since $|M_E| = |M^*| = \epsilon m$:

$$\mathrm{KL}(M_E||M^*) + \zeta\epsilon m = \epsilon m\,\mathrm{RE}(D_E||D^*) + \epsilon m \log(1) + 0 + \zeta\epsilon m$$
$$= \epsilon m\left[\mathrm{RE}(D_E||D^*) + \zeta\right]$$

Substituting the expansions back into (32) and dividing by $\epsilon m$, we get

$$\mathrm{RE}(D_E||D^{t+1}) \le \mathrm{RE}(D_E||D^*) + \log(1 + \zeta)$$

Using the inequality $\log(1 + \zeta) \le \zeta$, we obtain the first implication of the lemma:

$$\mathrm{RE}(D_E||D^{t+1}) \le \mathrm{RE}(D_E||D^*) + \zeta. \tag{33}$$

Next, we can apply Bregman's Theorem (Theorem 2.5) because $M^*$ is the exact projection of $N^{t+1}$ onto the convex set $\Gamma_\epsilon$, and $M_E \in \Gamma_\epsilon$:

$$\mathrm{KL}(M_E||N^{t+1}) \ge \mathrm{KL}(M_E||M^*) + \mathrm{KL}(M^*||N^{t+1}) \tag{34}$$

We expand all three terms using the KL-RE identity:

$$\mathrm{KL}(M_E||N^{t+1}) = |M_E|\,\mathrm{RE}(D_E||\hat{D}^{t+1}) + \underbrace{|M_E|\log\left(\frac{|M_E|}{|N^{t+1}|}\right) + |N^{t+1}| - |M_E|}_{T_M}$$

$$\mathrm{KL}(M_E||M^*) = \epsilon m\,\mathrm{RE}(D_E||D^*)$$

$$\mathrm{KL}(M^*||N^{t+1}) = |M^*|\,\mathrm{RE}(D^*||\hat{D}^{t+1}) + \underbrace{|M^*|\log\left(\frac{|M^*|}{|N^{t+1}|}\right) + |N^{t+1}| - |M^*|}_{T_M'}$$

Since $|M^*| = |M_E| = \epsilon m$, the terms $T_M$ and $T_M'$ are equal. Substituting the expansions back into the Bregman inequality (34) and dividing by $\epsilon m$ gives:

$$\mathrm{RE}(D_E||\hat{D}^{t+1}) \ge \mathrm{RE}(D_E||D^*) + \mathrm{RE}(D^*||\hat{D}^{t+1}) \ge \mathrm{RE}(D_E||D^*),$$

where the last inequality used the non-negativity of relative entropy. Combining with Equation (33), we obtain the second implication of the lemma

$$\mathrm{RE}(D_E||D^{t+1}) \le \mathrm{RE}(D_E||\hat{D}^{t+1}) + \zeta.$$

$$\square$$

