# OpenReview forum: "QuantumBoost: A lazy, yet fast, quantum algorithm for learning with weak hypotheses"
_ICML.cc/2026/Conference — ICML 2026 regular_

### Official Review · Reviewer_zZcH · 2026-03-10

**Soundness:** 3
**Presentation:** 3
**Significance:** 3
**Originality:** 3
**Overall Recommendation:** 4
**Confidence:** 3

**Summary:**

The authors propose a quantum version of a boosting algorithm. In the setting where we are given access to a weak learner and a dataset S, they can boost the learning algorithm to produce an hypothesis with small empirical error and a runtime of O(1/\sqrt{\epsilon}\gamma^4). One of the contributions is the quadratic improvement in the scaling of \epsilon, which seems to come from standard state preparation algorithms.

The second result explores the generalization abilities of the hypothesis generated by quantumboost, and Corollary 4.4 shows the size of the training set - sampled from a distribution D - needed to find an hypothesis H(x) that fails on new samples with probability $\leq \epsilon$.

**Compliance With Llm Reviewing Policy:**

Affirmed.

**Key Questions For Authors:**

- It seems to me, that the idea of not explicitly keep track of the m-dimensional weight vector m^t, but rather storing after each iteration the name of the weak hypothesis $h_t$ that was generated in that iteration, holds some similarity with the quantum simplex algorithm by Nannicini. That algorithm is also avoiding to perform full state tomography over quantum state but is only storing the index of quantum vector that is already known to the quantum algorithm in the memory. Is this intuition correct?

- It would be helpful to explain why the paper uses a definition of KL divergence which includes the additional terms $+N(x)-M(x)$. What I did not fully understand from a cursory reading. Is this the definition of KL divergence for non-probability measures?

- In the third point of "our improvements over earlier boosting algorithms"  they state that: "Even for the preparation of classical random examples which are needed for a classical weak learner, first preparing a quantum example and then measuring it is more efficient than classical rejection sampling". Can the authors clarify on this? How is this related to the overall speed up?

- What is the space complexity of the algorithm (width of the circuit)? Is it given by the number of copies of $\ket{D}$ (and a factor that depends on the learner)?

**Limitations:**

yes

**Strengths And Weaknesses:**

-  It would be worth to state more explicitly: was there any difficulty in working with measures on a quantum computer: it seems that quantum computers are naturally suited to work with probability distributions, but it is less clear to me how efficiently they can represent or manipulate more general measures.


- I anticipate that some reviewers would ask for experiments. Unlike them, I do not think experimental results are necessary for this work.

- In general I think that the quality and scope of the submission could be improved by working on some of the future works proposed in the  conclusion.


- I appreciate the technical results presented in this work. However, the runtime improvements seem to rely mainly on standard ingredients (e.g. mean estimation and binary search, as in Theorem 4.5) rather than on a substantially new algorithmic technique.  From my limited experience with  ICML, but I am not yet convinced that the level of novelty is sufficient for acceptance. However, I personally don't think this should be a reason for rejecting a paper, as most of the quantum algorithm works are using very similar subroutines.


- I appreciated the techniques used to prove the soundness of the algorithms in section 4.1 (MWU, bergman projections..)

---

> ### Author Rebuttal · Authors · 2026-03-30
>
> We thank the reviewer for their questions, suggestions and overall feedback which helps us improve the quality of our work. We respond to their points raised below:
>
> # Weaknesses
> 1. Working with measures: This was not a problem; the measures (weight vectors) are represented classically, while quantum states (examples for the weak learner) are prepared according to their normalization, which are probability distributions.
>
> 2. Working on some of the future works proposed in the conclusion: We agree. After the submission we in fact solved at least one of our open problems (obtaining a tight lower bound on the number of calls to the weak learner) and we will add that to the paper.
>
> 3. Runtime improvements: Our quantum tools are indeed fairly standard. The main novelties are (1) adapting the approximate-Bregman-projection approach of Kale to the quantum case, and (2) our "lazy" approach where we only do the  projections in about $\sqrt{T}$ of the $T$ iterations. To our knowledge, this is the first time this lazy technique (common in convex optimization) has been employed in boosting.
>
>
> # Questions
> 1.  Implicitly keeping track of $m^t$: Some implicit representation of the weight vector was also used in the first quantum boosting paper (Arunachalam and Maity, ICML'20). We don't do this to avoid quantum state tomography, but to avoid the linear-in-$m$ cost of explicitly updating the $m$-dimensional weight vector in each iteration.
>
> 2. KL-divergence: Yes, this is KL divergence for non-normalized measures, which is a crucial distinction for our proof. When restricted to probability distributions, the additional terms vanish, and KL divergence and relative entropy coincide.
>
> 3. Rejection sampling: One can prepare a classical example by preparing a quantum example (which takes time  roughly $1/\sqrt{\epsilon}$ for a smooth measure thanks to amplitude amplification) and then just measuring it. In contrast, classical rejection sampling to prepare an example is quadratically slower. However, if one explicitly keeps track of the $m$-dimensional vector of weights (which AdaBoost and SmoothBoost do, but the quantum boosters don't) then one can more efficiently prepare the examples for the weak learner.
>
> 4. Space complexity: The number of qubits needed is indeed determined by the number of qubits for the $W$ examples in each invocation of the weak learner. In addition we use $\text{polylog}(m)$ qubits for doing the approximate projection, and classical space (not QCRAM) proportional to $T=O(\log(1/\epsilon)/\gamma^2)$ for storing the weak hypothesis $h_t$ and the approximate-projection constant $c_t$ from each iteration.

---

> > ### Author Rebuttal · Reviewer_zZcH · 2026-04-03
> >
> > Thanks for your answers.

---

### Official Review · Reviewer_Ueke · 2026-03-11

**Soundness:** 3
**Presentation:** 2
**Significance:** 3
**Originality:** 2
**Overall Recommendation:** 4
**Confidence:** 3

**Summary:**

This work presents QuantumBoost, a quantum boosting algorithm with runtime $\tilde{O}(W / \sqrt{\epsilon}\,\gamma^4)$ where $W$ is the weak learner's runtime, $\epsilon$ the target error, and $\gamma$ the strength of the weak learner. Two key ingredients result in the improvement: (i) using quantum amplitude estimation to compute approximate Bregman projections faster, and (ii) a "lazy" projection strategy that, instead of projecting on every iteration, projects onto the set of high-density measures $\Gamma_\epsilon$ only every $K = 1/\gamma$ iterations. QuantumBoost matches (classical) AdaBoost's $T = O(\log(1/\epsilon)/\gamma^2)$ iteration count while removing explicit dependence on the training set size $m$.

**Compliance With Llm Reviewing Policy:**

Affirmed.

**Final Justification:**

I already had a positive score for this work. I had some confusions and the authors have clarified it in the rebuttal. I keep my original positive score.

**Key Questions For Authors:**

* The $\gamma^4$ dependence comes from the approximate Bregman projection cost. Can authors comment on whether or not this is a fundamental barrier, or could a different projection subroutine improve it to $\gamma^2$?
* The notion of "no explicit dependence on $m$ or $d$" for QuantumBoost is confusing, especially given that previous quantum boosting algorithms improve the dependence on $m$ (or $d$) at the expense of having worse dependence on $\gamma$. There is some discussion around this in Section 1.2, but some clarification would be helpful. Is this fundamental? Can it be traded off with the $\epsilon$ improvement?
* (minor) Proof of Theorem 4.6 ends with broken equation links.

**Limitations:**

Yes

**Strengths And Weaknesses:**

### Strengths
* Achieves $\tilde{O}(W/\sqrt{\epsilon}\,\gamma^4)$, improving on all prior quantum and classical boosters. Moreover, $1/\sqrt{\epsilon}$ dependence is new, as all previous methods had at least $1/\epsilon$. It would be great if the authors could provide more comment about why such improvement was possible.
* The idea of projecting on every $K$-th iteration (instead of every iteration) is a natural idea.
* Avoids explicit $m$-dimensional weight vectors updates by storing only the weak hypotheses $h_t$ and projection constants $\tilde{c}_t$, entries of the measure $M_t$ can be computed on-the-fly in $O(t)$ time.

### Weaknesses
* The paper acknowledges that $\gamma^4$ scaling is worse than classical AdaBoost's $\gamma^2$. For weak learners with small $\gamma$, this could dominate the runtime. This is acknowledged by the authors but I wonder if there is any room for improvement.
* Table 1 can be clarified. Since there are variants of "smooth boost" it is a bit confusing to compare different algorithms in the Table. The choice of suppressing polylog factors in the total runtime column and using big O notation in the third column seem inconsistent as well.
* The generalization bound does not improve over classical. Please refer to the questions below pertaining to this point.
* The algorithm requires quantum access to the training data (QCROM) and a quantum weak learner, which might be non-trivial to realize in practice.

---

> ### Author Rebuttal · Authors · 2026-03-30
>
> We thank the reviewer for their response to our work. We address their weaknesses and questions below, as well as point to relevant responses made to other reviewers where necessary.
>
> # Weaknesses
> - $\gamma^4$ scaling: AdaBoost also has a $\gamma^4$ scaling in its time complexity: it uses roughly $1/\gamma^2$ iterations, each of which has to update a vector of $m\geq  1/\gamma^2$ entries, thus resulting in a $1/\gamma^4$ scaling. Please also see our response to reviewer 56Ux explaining more on the $\gamma$ scaling.
>
> - Table 1: Fair point, thank you. We'll remove the $O$'s in Table 1.
>
> - QCROM: As mentioned in an earlier response to reviewer 56Ux, we will highlight this QCROM point more in the main text (rather than in the appendix). Our QuantumBoost strong learner works equally well with classical weak learners and quantum weak learners. Allowing the latter strengthens our results, but isn't necessary: one can just apply QuantumBoost on a classical weak learner.
>
> # Questions
> 1. Bregman projections: We don't know whether the $\gamma^4$ dependence in the overall runtime is optimal. However, we do know that at least $1/\gamma^2$ calls to the weak learner are needed for any (quantum or classical) boosting strategy, and all known boosting algorithms spend $1/\gamma^2$ (or more) work around each of those calls. Hence a $1/\gamma^4$ lower bound on the overall runtime is plausible, albeit hard to prove like all circuit lower bounds.
>
> 2. VC dimension dependence: This is a subtle point. Please see our responses to reviewer 56Ux.
>
> 3. Equation links: Thank you, we will fix this.

---

> > ### Author Rebuttal · Reviewer_Ueke · 2026-04-03
> >
> > The authors have address my concerns.

---

### Official Review · Reviewer_jFnQ · 2026-03-12

**Soundness:** 3
**Presentation:** 3
**Significance:** 3
**Originality:** 3
**Overall Recommendation:** 5
**Confidence:** 3

**Summary:**

This paper presents a quantum boosting algorithm, which boosts a $\gamma$-weak learner drawn from a hypothesis class of VC dimension $d$ (invoking which takes $W$ time), to achieve empirical error $\epsilon$ on a training dataset of $m$ examples, using $T=O(\log(1/\epsilon)/\gamma^2)$ iterations of boosting, and a running time of $O(W/\sqrt{\epsilon}\gamma^4)$. This improves previous guarantees on both fronts---number of iterations as well as running time. The paper shows how approximate Bregman projections used in previous work by Barak et al. 2009 may be performed efficiently by a quantum algorithm, and combines this with performing projections not at every iteration of boosting, but only at every $K^{\text{th}}$ iteration.

**Compliance With Llm Reviewing Policy:**

Affirmed.

**Final Justification:**

The author rebuttal suitably addresses my questions, and I maintain my positive evaluation of this paper

**Key Questions For Authors:**

1) To my knowledge, the current best sample complexity for AdaBoost is derived in https://arxiv.org/pdf/2502.16462v2, and has a $d\log(\cdot)\log^2(\cdot)/\gamma^2\epsilon$ dependence, which is tight upto the $\log^2(\cdot)$ dependence. How does the sample complexity bound in Theorem 1.2 compare to this? Is there a lower bound on the optimal sample complexity that can be achieved by a quantum boosting algorithm?

2) It is a little surprising to me that the running time has no dependence on the VC dimension $d$ of the weak learner class. Could the authors elaborate on why this is the case, at a high level? How is the algorithm able to get rid of this dependence?

**Limitations:**

yes

**Strengths And Weaknesses:**

### Strengths:

The results in the paper concretely improve upon the state of the art. The paper is well-written and the technical contributions are contextualized well within prior literature. The high-level techniques are described accessibly and can be of broader interest.

### Weaknesses:

It was not immediately clear to me what is a lower bound on the optimal (classical/quantum) running time of a boosting algorithm in this context. The authors could clearly state what these are in Table 1 as well, along with lower bounds on the optimal number of iterations required to achieve a given empirical error.

---

> ### Author Rebuttal · Authors · 2026-03-30
>
> We thank the reviewer for their positive feedback regarding our work and have responded to their questions below:
>
> 1. Runtime lower bounds: We know that at least $1/\gamma^2$ calls to the weak learner are needed for any (quantum or classical) boosting strategy, and all known boosting algorithms spend $1/\gamma^2$ (or more) work around each of those calls. Hence a $1/\gamma^4$ lower bound on the overall runtime is plausible, albeit hard to prove like all circuit lower bounds.
>
>      More precise bounds have been proven for sample complexity. Blanc, Hayderi, Koch, Tan [arXiv:2409.11597] prove that $\tilde{\Omega}(w/\gamma^2)$ samples are necessary for smooth boosting strategies (including ours), where $w$ is the size of the  training set given to the weak learner. Similarly, Green Larsen and Ritzert [arXiv:2206.01563] give a sample complexity lower bound for boosting strategies like Adaboost. We will add these references and results on the sample complexity lower bounds to the main text and highlight them. We'd rather not add those to Table 1 itself, since that table focuses only on time complexity upper bounds.
>
>
> 2. Sample complexity: The optimal sample complexity for general boosting algorithms is $\tilde{\Theta}(d/(\gamma^2\epsilon))$ [Green Larsen and Ritzert, arXiv:2206.01563]. The sample complexity of our quantum booster in Theorem 1.2 is $\tilde{O}(d/(\gamma^2\epsilon^2))$, which has an extra $1/\epsilon$ factor. However, this matches the best known sample complexity $O(d/(\gamma^2\epsilon^2))$ among all known _smooth_ boosting algorithms, and is optimal up to logarithmic factors given the corresponding lower bound $\tilde{\Omega}(d/(\gamma^2\epsilon^2))$ [arXiv:2409.11597]. Our main contribution lies in the improved time complexity (the amount of work done around each call to the weak learner is only polylogaritmic in $m$, not polynomial like all earlier classical and quantum boosting algorithms), not the sample complexity.
>
> 3. VC dimension: We do not have a polynomial dependence on the number $m$ of examples (which is linear in the VC-dimension of the weak learner's hypothesis class) because we only _implicitly_ represent the $m$-dimensional weight-vector, and because the quantum-counting approximations needed to do an approximate Bregman projection only need a polylogarithmic number of queries to that weight-vector. There is, however, still an implicit dependence on the VC-dimension because of the runtime $W$ of the weak learner, which depends on the VC-dimension. We explain why this is the case in lines 147-156 of the submission and in our reply to reviewer 56Ux, and we will clarify this further in the paper.

---

> > ### Author Rebuttal · Reviewer_jFnQ · 2026-04-03
> >
> > Thank you for the response. I will maintain my positive score. It would be great if the discussion above is suitably referenced in the paper.

---

### Official Review · Reviewer_56Ux · 2026-03-13

**Soundness:** 2
**Presentation:** 3
**Significance:** 2
**Originality:** 2
**Overall Recommendation:** 4
**Confidence:** 4

**Summary:**

The manuscript proposes QuantumBoost, a quantum algorithm for boosting weak learners into strong learners, inspired by the classical SmoothBoost algorithm by Barak, Hardt, and Kale. The authors aim to improve the runtime of existing classical and quantum boosting methods through two primary innovations: utilizing a quantum mean estimation subroutine to compute approximate Bregman projections faster, and introducing a "lazy projection" strategy where the high-density constraint is only enforced every $K$ iterations rather than at every step. The authors claim an overall runtime of $\tilde{O}(\frac{W}{\sqrt{\epsilon}\gamma^4})$, which notably improves the dependence on the empirical error $\epsilon$ and allegedly removes the explicit runtime dependence on the training set size $m$ and the VC-dimension $d$.

**Compliance With Llm Reviewing Policy:**

Affirmed.

**Final Justification:**

The rebuttal addressed my mian concerns.

**Key Questions For Authors:**

Here are some questions for authors:

1. To provide a fair comparison in Table 1, what does the true end-to-end complexity of QuantumBoost look like when $W$ is explicitly expanded in terms of the VC-dimension $d_{\mathcal{C}}$ and the weak learner advantage $\gamma$?
2. Given the $1/\gamma^4$ scaling, is there a theoretical threshold for $\gamma$ at which QuantumBoost strictly loses its asymptotic advantage over classical AdaBoost?
3. The state preparation subroutine relies heavily on QCROM. How would the algorithm's runtime scale if the QCROM assumption is relaxed to a more realistic, near-term data-loading scheme (e.g., using explicit quantum circuits to load classical data)?

**Limitations:**

No, the authors have not adequately discussed the limitations and potential negative societal impact of their work.

Constructive Suggestions for Improvement:
1. The authors relegate a highly critical limitation—the reliance on the controversial QCROM memory model—to Appendix B. Because the algorithm's entire speedup hinges on this assumption, this discussion must be elevated to a dedicated "Limitations" section in the main text.
2. The current impact statement is dismissive, claiming there are "no societal consequences" until large-scale quantum computers arrive. The authors should replace this with a substantive reflection on the potential consequences of their work. For instance, they could discuss the dual-use nature of exponentially accelerated machine learning frameworks or acknowledge the immense resource footprint and energy costs required to maintain coherent QCROM queries over large datasets.

**Strengths And Weaknesses:**

Strengths:
1. Adapting the lazy projection strategy from convex optimization into a quantum boosting framework is a mathematically elegant and novel approach. By only projecting every $K=1/\gamma$ iterations, the authors successfully reduce the average per-iteration overhead.
2. The algorithm is the first boosting method (classical or quantum) to successfully reduce the runtime dependence on the empirical error from $O(1/\epsilon)$ to $O(1/\sqrt{\epsilon})$.

Weaknesses:
1. The claim in Table 1 that QuantumBoost removes explicit dependence on the VC-dimension $d$ is highly misleading. In Section 1.2, the authors concede that the weak learner's sample complexity, and thus its runtime $W$, is intrinsically lower-bounded by $\gamma^2 d_{\mathcal{C}}$. Therefore, the $d$ dependence is not eliminated. It is merely hidden inside the $W$ variable, making the comparative runtime claims appear more favorable than they actually are.
2. The proposed runtime scales with $1/\gamma^4$. In contrast, classical AdaBoost scales with $1/\gamma^2$ for the $W$ term. Since $\gamma$ represents the weak learner's slight advantage over random guessing (which is typically very small), raising it to the fourth power poses a severe bottleneck that could easily negate any theoretical quantum speedup gained elsewhere.
3. The algorithm's efficiency relies entirely on the existence of a Quantum Read-Only Memory (QCROM) to query classical training examples in superposition in $O(1)$ time. As the authors briefly admit in Appendix B, QCROM is a "controversial notion" because it is exceptionally difficult to implement in practice without massive overhead on actual physical hardware.

---

> ### Author Rebuttal · Authors · 2026-03-30
>
> We thank the reviewer for their insightful comments, which help us clarify some subtle points in our paper. The three weaknesses that the reviewer mentions correspond to their three subsequent questions, which we respond to below.
>
> # Questions
>
> 1. VC dimension dependence: It is a crucial and subtle point that the time complexity of our QuantumBoost algorithm implicitly still depends on the VC-dimension, even though this is not explicit in our Table 1. The reason, as we explained after the table (lines 147-156 of the submission), is that the sample complexity of the weak learner (and hence its time complexity $W$) is at least $\gamma^2 d_C$, where $d_C$ is the VC-dimension of the target class $\cal C$. We cannot rewrite the upper bounds on time complexity of Table 1 in terms of $d_C$ and $\gamma$, because we do not have an upper bound on $W$ in terms of $d_C$ and $\gamma$, only the mentioned lower bound (and the time complexity $W$ of a learning algorithm can of course be _much_ larger than its sample complexity). Boosting algorithms have to be able to work with any weak learner that can produce $(1/2+\gamma)$-weak hypotheses, and thus we leave $W$ as a placeholder here for the weak learner's time complexity, and also as an upper bound for its sample complexity. The same was done in the earlier papers about quantum boosting. We will clarify this in the paper.
>
> 2. The $1/\gamma^4$ scaling: Referring to the last part of the previous response; we could complicate notation by having a separate symbol, say $w$, for the sample complexity of the weak learner, and continue to use $W$ for its runtime (potentially $w\ll W$). In that case QuantumBoost's runtime upper bound would be $\displaystyle\frac{W}{\gamma^2}+\frac{w}{\sqrt{\epsilon}\gamma^4}$ and AdaBoost's would be $\displaystyle\frac{W}{\gamma^2}+\frac{m+w}{\gamma^2}$ (up to log-factors). Both QuantumBoost and AdaBoost use $1/\gamma^2$ iterations, each of which invoke the weak learner once; after the submission of our paper we found that the classical lower bound of $\Omega(1/\gamma^2)$ on the number of calls to the weak learner also holds for quantum boosters, so the $W/\gamma^2$ dependence is best-possible. Also, both QuantumBoost and AdaBoost have an overall runtime that's linear in $1/\gamma^4$. In the case of AdaBoost this is because in each of the $1/\gamma^2$ iterations it updates each of the weights of the $m$ examples, and $m$ is linear in $1/\gamma^2$. In the case of QuantumBoost this is because in each of the $1/\gamma^2$ iterations it generates $w$ examples for the weak learner, and each example-generation has a cost that is linear in $T$, and hence linear in $1/\gamma^2$.
> The $\epsilon$-dependence of QuantumBoost is significantly better than AdaBoost's because the latter's sample size $m$ is at least $d_W/\epsilon\gamma^2$, where $d_W$ is the VC-dimension of the weak learner's hypothesis class (which we call $d$ in the submission).
>
>     __To answer the question:__ AdaBoost is faster than QuantumBoost if its overall sample complexity $m$ satisfies $m+w\ll w/\sqrt{\epsilon}\gamma^2$ which we will add to the paper for further clarification.
>
> 3. QCROM: Implementing QCROM for an $N$-bit memory can actually be done with a $\log(N)$-depth circuit (arXiv:2406.18030) of size roughly $N$. Since our computational model counts circuit size, factoring in the circuit-size cost of the QCROM queries to the $m$-element input sample would increase our cost  by a rather expensive factor of $m \sim d_W/\epsilon^2\gamma^2$, which would remove any speedup. However, this is not quite a fair comparison: classical RAM requires similar linear circuit size as QCROM and yet classical RAM-queries are generally not considered expensive. Conceptually, QCROM is a natural combination of RAM and quantum superposition, and we feel it is an appropriate resource for the theory track of ICML.
>
>
> # Suggestions
>
> 1. About QCROM: We agree, we will move this important point from the appendix to the main text in order to highlight it.
> QCROM is not the only thing that is responsible for our speedup, though: several quantum-algorithmic techniques and especially our lazy strategy of doing approximate Bregman projections only in some of the iterations are also essential.
> 2. Impact statement: We believe that this theoretical research really has rather few societal consequences, but the reviewer raises good points and we will add text about the resource footprint and energy costs in line with what was suggested by the reviewer.
>
> Again, we thank the reviewer for their insightful feedback. This has really helped us determine points which we will further clarify.

---

> > ### Author Rebuttal · Reviewer_56Ux · 2026-04-04
> >
> > Thanks for the authors. I will keep my score.

---

> > > ### Author Response · Authors · 2026-04-07
> > >
> > > We thank the reviewer for acknowledging our rebuttal and kindly ask if they could explain why they have decided to keep their score in light of marking their concerns as fully addressed. This would be helpful for our understanding, especially since all other reviewers have marked their concerns fully addressed, yet had more positive scores.

---

### Decision · Program_Chairs · 2026-04-30

**Decision:**

Accept (regular)

**Comment:**

Reviewers agreed that the contribution is original and elegant. Some initial concerns have been raised about the novelty level, but as quantum machine learning is still an emergent topic, my assessment is that merging ideas to improve previous results is very relevant to the field.

Following the diligent reviews, the authors have clarified important aspects of their work, notably regarding the complexity of their algorithm.  Therefore, I enjoin the authors to carefully incorporate these remarks into the final version of their manuscript.